# In-Composition Graded Quantum Barriers for Polarization Manipulation in InGaN-Based Yellow Light-Emitting Diodes

**DOI:** 10.3390/ma15238649

**Published:** 2022-12-04

**Authors:** Siyuan Cui, Guoyi Tao, Liyan Gong, Xiaoyu Zhao, Shengjun Zhou

**Affiliations:** 1Center for Photonics and Semiconductors, School of Power and Mechanical Engineering, Wuhan University, Wuhan 430072, China; 2The Institute of Technological Sciences, Wuhan University, Wuhan 430072, China

**Keywords:** indium gallium nitride, yellow LED, quantum barrier structure, optoelectronic device

## Abstract

Highly efficient indium gallium nitride (InGaN)-based yellow light-emitting diodes (LEDs) with low efficiency droop have always been pursued for next-generation displays and lighting products. In this work, we report an InGaN quantum barrier (QB) with linear-increase In-composition along [0001] direction for InGaN-based yellow LEDs. With the In-composition in QBs systematically engineered, three QB structures including linear-increase QB (LIQB), linear-decrease QB (LDQB) and commonly used flat QB (FQB) were investigated by simulation. The results show that the LIQB not only yields enhanced electron confinement, but also contributes to suppressed polarization field. Consequently, the yellow LED incorporated with LIQBs demonstrates improved radiative recombination rates and the efficiency droop is alleviated. Under a current density of 100 A/cm^2^, the efficiency droop ratios of LEDs with FQBs, LDQBs and LIQBs are 58.7%, 62.2% and 51.5%, respectively. When current density varies from 1 A/cm^2^ to 60 A/cm^2^, the blueshift values of peak emission wavelength for LEDs with FQBs, LDQBs and LIQBs are 14.4 nm, 16.5 nm and 13.0 nm, respectively. This work is believed to provide a feasible solution for high-performance InGaN-based LEDs in long-wavelength spectral region.

## 1. Introduction

Indium gallium nitride (InGaN)-based light-emitting diodes (LEDs) have developed over decades and demonstrated extensive potential in solid-state lighting and high-resolution displays [1,2,3,4,5,6]. The wide emission spectrum from ultraviolet to infrared can be achieved through modulating the In composition in multiple quantum wells (MQWs). Moreover, the color mixing of red/green/blue emitters is seen to be promising for high-resolution full-color displays [7,8,9]. However, there is still a challenge to obtain efficient LEDs emitting in green-yellow region. While the efficiency of blue and red LEDs exceeds 80% and 50%, respectively, InGaN-based yellow LEDs exhibit unsatisfactory quantum efficiency [10,11,12]. This phenomenon is commonly referred to be the “green-yellow gap” and limits the large-scale application of yellow LEDs [13]. The increment of In concentration in In_x_Ga_1−x_N quantum well (QW) in yellow LEDs can be involved with this efficiency gap [14]. With the high In-content in QWs, lattice mismatch inevitably becomes severe between the InGaN QW layer and GaN quantum barrier (QB). This eventually gives rise to internal electrostatic fields and a quantum confined Stark effect (QCSE) along the growth direction [15,16]. The optoelectronic performance of yellow LEDs degenerates for the reason that the QCSE brings a drop in electron-hole wavefunction overlap, further deteriorating the radiative recombination rate in yellow LEDs [17]. More importantly, the internal quantum efficiency (IQE) of LEDs decreases drastically as the operating current density increases [18]. This problem is the so-called “efficiency droop”, which is a crucial issue for InGaN-based LEDs, especially yellow LEDs [19]. The separation of carrier wavefunction arising from strong polarization effects is supposedly one of the main causes, although it is not the only explanation [20,21].

Many approaches focusing on polarization manipulation have been proven effective in addressing the QCSE, and the efficiency droop ratio can be reduced. For example, some numerical analyses have proved that novel MQW structures, such as graded QWs [22], staggered QWs [23] and triangular QWs [24], are effective in boosting the luminous efficiency. Because of this, the detrimental impact of polarization mismatch, that is, the poor carrier injection into MQWs, can be countered to some extent. In experiments, to relieve the in-plane residual stress and improve the crystal quality of InGaN alloys, semipolar and nonpolar GaN films are selected as the substrates on which InGaN/GaN MQWs are grown [25]. The reasonably high cost of these materials cannot be ignored, despite the low density of defects and advanced device performance [26]. Furthermore, the parameters of QB also serve a major role in electric field modulation, carrier distribution and crystal quality across the MQWs [27,28]. Nevertheless, efforts to the effect of QB modifications on yellow emission are currently inadequate and therefore the corresponding mechanisms are worth further investigation.

Here, we have proposed a bandgap-engineering strategy to alleviate the polarization effect for InGaN-based yellow LEDs. By adopting an In-composition linear-increase QB (LIQB) in yellow LEDs, the electron leakage is significantly reduced compared to LEDs with In-composition linear-decrease QB (LDQB) and flat QB (FQB). The specially designed LIQB structure can effectively mitigate the QCSE induced from the polarization of electric fields. As a result, the yellow LED with LIQBs earns facilitated radiative efficiency and bears a less severe efficiency droop. Moreover, in the yellow LED with LIQBs, a relatively moderate blueshift of peak emission wavelength is observed.

## 2. Device Structures and Parameters

The epilayers of the yellow LED with FQBs (denoted as sample A) are composed of 1.5-μm-thick n-GaN (Si: 5 × 10^18^ cm^−3^), nine periods of In_0.35_Ga_0.65_N (3 nm)/In_0.03_Ga_0.97_N (12 nm) MQWs active region, an electron-blocking layer (EBL) of 40-nm-thick p-Al_0.2_Ga_0.8_N (Mg: 2 × 10^19^ cm^−3^) and a of 200-nm-thick p-GaN (Mg: 5 × 10^19^ cm^−3^). The device structure of an LED with LDQBs (denoted as sample B) and sample A are identical except for each QB, which has linear-decrease In-composition from 0.06 to 0 along [0001] direction. Similarly, the LED with LIQBs (denoted as sample C) possesses an identical epitaxial structure other than nine InGaN QBs. The In-composition of QBs in sample C increases linearly from 0 to 0.06 along [0001] direction. Figure 1 is the schematic representation for the yellow LED epitaxial structures of the three samples. Numerical calculations were conducted by the SiLENSe version 5.14 software [29] to analyze the device properties of LEDs with three different QB structures. This software is implemented with a one-dimensional drift-diffusion transport model and takes the Fermi-Dirac statistics, Poisson equation and Schrödinger equations into account. In the simulation procedure, the material parameters at 300 K for wurtzite nitride semiconductors are summarized in Table 1. The electrons and holes mobility are set as 100 cm^2^V^−1^s^−1^ and 10 cm^2^V^−1^s^−1^, respectively. The energy band offset ratio between the conduction band and valence band is 70/30 [30]. The non-radiative recombination lifetime is assumed to be 1 ns and the ratio of recombination to non-radiative recombination lifetime determines the IQE of LEDs [31,32,33]. The dislocation density is considered to be 8 × 10^8^ cm^−2^. The device geometry is designed as 300 × 300 μm^2^.

## 3. Results and Discussion

The carriers’ injection and transport across the MQWs region are associated with an effective potential barrier at the interface of the last QB/EBL. Figure 2a–c illustrates the energy band diagrams of sample A, sample B and sample C near the MQWs region at 60 A/cm^2^. In Figure 2a,b, the effective potential barrier heights of electrons are 598 meV and 578 meV in sample A and sample B, respectively. When LIQBs in sample C are employed, the effective barrier height of electrons is further increased to 614 meV. This increased conduction barrier height in sample C enables an enhanced electron blocking capability, which can prevent electrons from overflowing out of MQWs without undergoing the radiative recombination process. Furthermore, the polarization charge induced at the interface of last QB/EBL bends the energy band downward. The distorted conduction band provides barrier heights for electrons, thereby helping to mitigate the electron leakage effect. On the other hand, the valence barrier height at the heterointerface of last QB and EBL also hinders holes from injecting into MQWs. As Figure 2a,b shows, the effective valence barrier height in sample B is reduced from 446 meV to 415 meV, when the FQBs of sample A are replaced by LDQBs. It is worth noting that the effective barrier height for holes is found to be 458 meV in sample C from Figure 2c. The reduced value of effective valence barrier height in sample B definitely contributes to the hole transport and distribution in active region. As a consequence, the optimum hole injection efficiency may appear in sample B among the three samples.

To further investigate the main function of these three QB structures, electron and hole concentration distribution, radiative recombination rate and electric fields in MQWs have been presented in Figure 3. As indicated in Figure 3a, the electron concentration of sample C is improved as the LIQB structure is applied. This is attributed to the increased barrier height of sample C, which matches well with our previous analysis. From Figure 3b, holes are observed to accumulate in the last QW owing to their high effective mass and low mobility. Furthermore, an enhancement of hole concentration in sample B can be obtained, as a comparison to the hole concentration of sample A and sample C. As discussed earlier, holes can inject into MQWs more efficiently, benefiting from the reduced valence barrier height, which promotes the hole concentration in MQWs. In Figure 3c, the radiative recombination rate in MQWs of the three samples is demonstrated at the current density of 60 A/cm^2^. The inset of Figure 3c exhibits the radiative recombination rate in the eight QWs of the three samples for better comparison. Interestingly, the radiative recombination rate of sample C is remarkably increased in all QWs except the last QW. Moreover, the radiative recombination rate of three samples is approximately the same in the last QW. The slight rise of electron concentration in sample C seems not to be enough to result in the increased radiative recombination rate. Moreover, the hole concentration of sample C is inferior to that of sample A and sample B.

The electric field profile in MQWs is illustrated in Figure 3d to reveal the underlying mechanism of the enhanced radiative recombination rate of sample C. It is well known that the electric field induced by strong polarization is closely correlated to the notorious QCSE. A severe spatial separation of the electron-hole wavefunctions caused by the QCSE brings down the radiative recombination rate in MQWs region. In order to investigate the polarization effect intrinsically, the net polarization charge density (P), electric field in QB ($E_{b}$) and electric field in QW ($E_{w}$) are mathematically expressed by the following equations [16,27]:(1)$$\Delta P=\sigma_{s1}^{Pol}{/}_{z=0}-\rho_{B}^{Pol}\cdot z\;(z<l_{b})$$
(2)$$\rho_{B}^{Pol}=\nabla \cdot P\left(z\right)=\left(\frac{\partial P}{\partial x}\right)\times \left(\frac{\partial x}{\partial z}\right)$$
(3)$$E_{b}\approx \frac{l_{w}\cdot \Delta P}{l_{b}\cdot \varepsilon_{w}+l_{w}\cdot \varepsilon_{b}}$$
(4)$$E_{b}\cdot l_{b}=E_{w}\cdot l_{w}$$
where $\sigma_{s1}^{Pol}$ is the polarization-induced sheet charge density and $\rho_{B}^{Pol}$ is the polarization-induced bulk charge density. x is the In composition in QBs. $l_{w}$ and $l_{b}$ are the QW and QB thickness. $\varepsilon_{w}$ and $\varepsilon_{b}$ are dielectric constants of QW and QB, respectively. The value of ∂P/∂x is a constant when considering the crystal relaxation level. Therefore, $\rho_{B}^{Pol}$ is as a function of ∂x/∂z. Polarization-induced bulk charges emerge in sample B and sample C owing to the linearly varying configuration of QBs. The ΔP for sample C is reduced because of the positive $\rho_{B}^{Pol}$ value whereas the ΔP for sample B is increased owing to the negative value of $\rho_{B}^{Pol}$. According to the Equations (2) and (3), we can safely deduce that the polarization-induced electric field is alleviated in sample C but that the polarization effect of sample B becomes even more severe. This phenomenon is obviously indicated in Figure 3d. From Figure 3d, it can be observed that sample C shows the weakest electric fields in MQWs when compared to sample A and sample B. Thanks to the polarization-induced bulk charge in LIQBs, the polarization electric field is manipulated, and thus QCSE is self-screened. In general, reduced internal electric field in MQWs can contribute to the increasing electron-hole wavefunction. The calculated electron-hole wavefunction overlap integrals in each QW for sample A, sample B and sample C are summarized in Table 2. As can be seen, the carrier wavefunction overlap of sample C is the highest among the three samples due to the mitigation of QCSE. As a result, the radiative recombination rate of sample C is substantially boosted, although a considerable improvement in carrier concentration is not attained in sample C. Hence, the overall performance of sample C exceeds sample A and sample B by taking advantage of LIQBs.

The IQE curves of three samples are provided in Figure 4. As shown in Figure 4, the maximum IQE values of sample A, sample B and sample C are 73.2%, 70.1% and 76.8%, respectively. This is consistent with the radiative recombination rate in the MQWs of the three samples. More significantly, at 100 A/cm^2^, the efficiency droop ratios of sample A, sample B and sample C are 58.7%, 62.2% and 51.5%, respectively. Based on our previous discussions, the excellent performance in efficiency droop of sample C is mainly ascribed to the enhanced radiative efficiency. Among the three sets of LED samples, sample C with LIQBs has the highest radiative recombination rate in MQWs. The suppressed QCSE in sample C definitely contributes to the overlap probability of electrons and holes. However, the strong polarization electric field in sample B aggravates the QCSE, which leads to the worst droop behavior among three samples, as shown in Figure 4.

Figure 5 exhibits the emission spectrum of three samples. As presented in Figure 5, three samples all emit in the yellow spectral region under 1 A/cm^2^ current density. The peak wavelengths of the three samples shift to the shorter region with increasing current density. When the current density increases from 1 A/cm^2^ to 60 A/cm^2^, the blueshift values of peak emission wavelength for sample A, sample B and sample C are 14.4 nm, 16.5 nm and 13.0 nm, respectively. The polarization electric field in MQWs can be reflected by the blueshift of peak emission wavelength [34]. Owing to the alleviated polarization effect and partly self-screened QCSE, sample C possesses the minimum value of blueshift among the three samples, which further confirms our aforementioned analysis.

## 4. Conclusions

To summarize, LIQBs were proposed and numerically investigated for highly efficient InGaN-based yellow LEDs. The device characteristics, including energy band, carrier concentration, radiative recombination rate, electric field, IQE and emission spectrum, have been extensively investigated in simulation. As compared to LEDs with FQBs and LIQBs, the yellow LED with LIQBs demonstrates boosted electron concentration and radiative recombination rates. Our analysis indicates that the polarization effect and QCSE can be eliminated by utilizing the LIQB structure. Therefore, the efficiency droop is less severe, and the blueshift of peak wavelength becomes relatively moderate in LEDs with LIQBs. Our unique LIQB design is suggested to provide a new insight into InGaN-based LEDs emitting in long wavelength.

## Figures and Tables

**Figure 1 materials-15-08649-f001:**
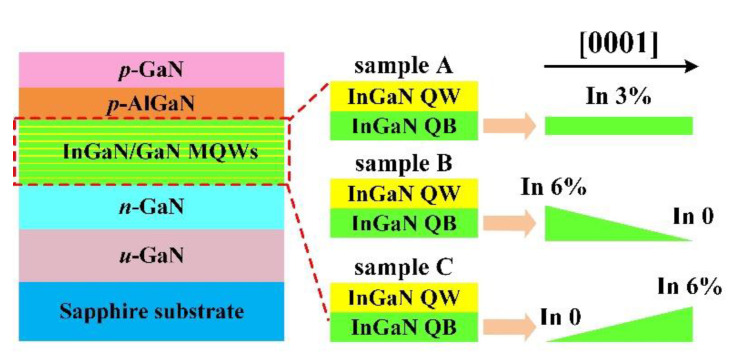
Schematic representation of the yellow LED epitaxial structures of sample A, sample B and sample C.

**Figure 2 materials-15-08649-f002:**
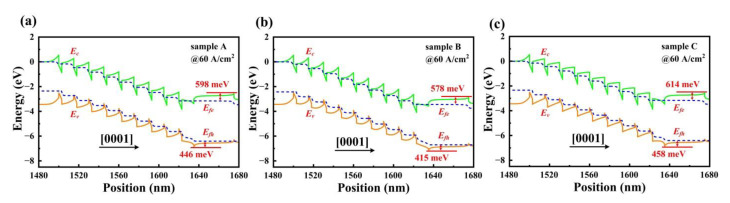
Energy band diagrams of (**a**) sample A, (**b**) sample B and (**c**) sample C near the MQWs region at current density of 60 A/cm^2^. E_fe_, E_fh_, E_v_ and E_c_ represent electron quasi-Fermi level, hole quasi-Fermi level, valence band and conduction band, respectively.

**Figure 3 materials-15-08649-f003:**
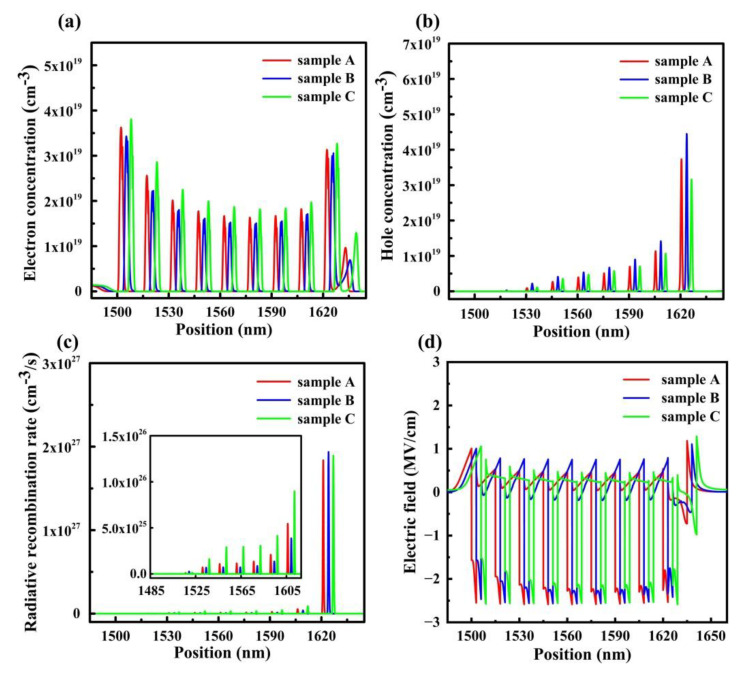
(**a**) Electron and (**b**) hole concentration distribution in MQWs for sample A, sample B and sample C. (**c**) Radiative recombination rate in MQWs for sample A, sample B and sample C. the inset shows the radiative recombination rate in eight QWs except the last QW. (**d**) Electric fields in MQWs for sample A, sample B and sample C. The data are acquired at 60 A/cm^2^. The electron concentration, hole concentration, radiative recombination rate and electric field profiles are purposely shifted by 3 nm to present the curves more clearly.

**Figure 4 materials-15-08649-f004:**
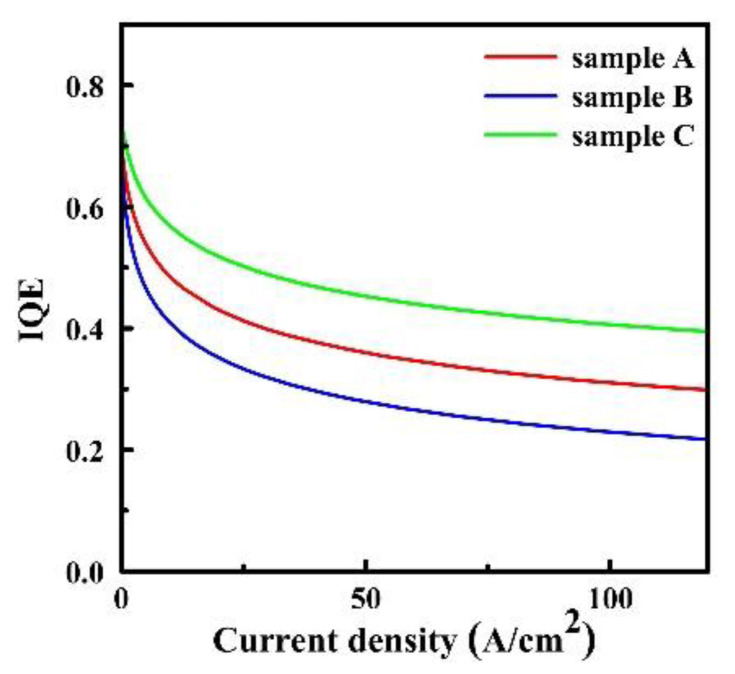
IQE with respect to current density of sample A, sample B and sample C.

**Figure 5 materials-15-08649-f005:**
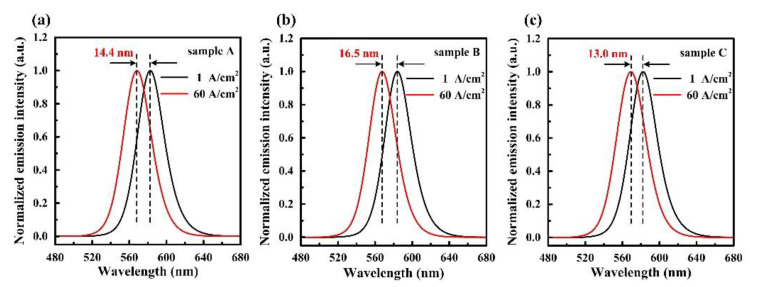
Normalized emission intensity of (**a**) sample A, (**b**) sample B and (**c**) sample C with increasing current density.

**Table 1 materials-15-08649-t001:** Material parameters at 300 K for wurtzite nitride semiconductors.

Parameters	Symbol (Unit)	GaN	AlN	InN
Lattice constant	*a_0_*(Å)	3.189	3.112	3.545
Bandgap energy	E*_g_*(eV)	3.435	6.138	0.711
Spin-orbit splitting	Δ*_so_*(eV)	0.017	0.019	0.005
Crystal-field splitting	Δ*_cr_*(eV)	0.010	−0.169	0.040
Elastic constant	*c_33_*(GPa)	398	373	224
	*c_13_*(GPa)	106	108	92
Piezoelectric coefficient	*d_33_*(pm/V)	3.1	5.4	7.6
	*d_13_*(pm/V)	−1.6	−2.1	−3.5
Spontaneous coefficient	*P_sp_*(C/m^2^)	−0.034	−0.09	−0.042

**Table 2 materials-15-08649-t002:** Wavefunction overlap in each QW for sample A, sample B and sample C.

	QW1	QW2	QW3	QW4	QW5	QW6	QW7	QW8	QW9
Sample A	22.0%	15.8%	13.6%	12.8%	12.3%	12.3%	12.4%	13.0%	17.1%
Sample B	21.3%	14.2%	12.5%	11.8%	11.5%	11.4%	11.7%	12.3%	17.3%
Sample C	22.9%	16.8%	14.3%	13.4%	13.0%	12.9%	12.9%	13.5%	17.2%

## Data Availability

Data are contained within the article.

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
