# Peer review of "In-Composition Graded Quantum Barriers for Polarization Manipulation in InGaN-Based Yellow Light-Emitting Diodes"

_materials, 2022, doi:10.3390/ma15238649_

Round 1

Reviewer 1 Report

In this manuscript, the authors simulated the effect of In-composition-graded quantum barriers on the performances of InGaN-based yellow light-emitting diodes. The authors' proposal would be interest of the researchers in the field of III-nitride-based optical devices. However, the manuscript includes some parts that should be revised and clarified before publication. The authors have to address the following questions and comments and modify their manuscript accordingly;

#1 (Details of the simulation conditions)

Please clarify the following information

- Version of SiLENSe

(*There are significant differences in the calculated results between ver. 6 and ver. 5 or older.)

- Are the well and barrier layers Si-doped or UID?

- AlN molar fraction of p-AlGaN EBL

- Lattice strain and relaxation of each layers

#2 (Figure 1)

In the right most of Figure 1, the authors schematically illustrate the gradient of the Indium composition of the barrier layers. I suggest that indicating the n-side and p-side in the figure will improve the readability.

#3 (Figure 3 and related discussion)

The authors state that the performance improvement in sample C is mainly owing to the mitigation of the QCSE and resultant enhancement of radiative recombination. Thus, the authors should discuss the electric field of quantum well layer (E-well) in more detail but the quantum barrier layer (E-barrier) because QCSE is related to E-well much stronger than E-barrier. In Figure 3(d), we can find that E-barrier is obviously modified by the Indium composition profiles of the barrier layers. However, it is unclear whether there are any differences in E-well among the three samples. Please modify (or add another graph) to make the difference in E-well clear.

#4 (Figure 3 and related discussion)

To prove the enhancement of the radiative recombination, indicating the square overlap integrals of electron and hole wavefunction is the most simple and easiest to understand. I recommend the authors to add the calculated data of overlap integrals.

#5 (Figure 4 and related discussion)

The unnormalized IQE curve should be shown together to show the actual performances of three devices.

Author Response

Thank you very much for reviewing our submitted paper and for your valuable suggestions and comments. These suggestions and comments are very conducive and helpful for improving our manuscript. We have complied with the reviewer’s suggestions to make a necessary revision and edit the entire manuscript (materials-2042235). 

Reviewer 2 Report

1. Please state the assumption of simulation to cater the non-idea setup.  If SiLENSe software was used for the proposed work, please cite.

2. In comparison to other equations, Equation-2 uses Cross (X). Please clarify.

3. If FQB was used as reference, the result could refer to FQB or normalized to FQB.

Author Response

(The authors gave the same response as above.)
